# Isolation and characterization of bovine herpes virus 5 (BoHV5) from cattle in India

**Naveen Kumar** ☉*, **Yogesh Chander**☉, **Thachamvally Riyesh**☉, **Nitin Khandelwal**, **Ram Kumar, Harish Kumar, Bhupendra N. Tripathi***, **Sanjay Barua***

National Centre for Veterinary Type Cultures, ICAR-National Research Centre on Equines, Hisar, India

☉ These authors contributed equally to this work.
* naveenkumar.icar@gmail.com (NK); sbarua06@gmail.com (SB); bntripath1@yahoo.co.in (BNT)

## Abstract

Bovine herpesvirus 1 (BoHV1) and 5 (BoHV5) are genetically and antigenically related alphaherpesviruses. Infection with one virus induces protective immunity against the other. However, disease associated with BoHV1 and BoHV5 varies significantly; whereas BoHV1 infection is usually associated with rhinotracheitis and abortion, BoHV5 causes encephalitis in cattle. BoHV5 outbreaks are sporadic and mainly restricted to the South American countries. We report BoHV5 infection for the first time from aborted cattle in India. Based on the characteristic cytopathic effects in MDBK cells, amplification of the viral genome by PCR, differential PCR for BoHV1/BoHV5, nucleotide sequencing and restriction endonuclease patterns, identity of the virus was confirmed as BoHV5 subtype A. Serum samples from the aborted cattle strongly neutralized both BoHV1 and BoHV5 suggesting an active viral infection in the herd. Upon *UL27*, *UL44* and *UL54* gene-based sequence and phylogenetic analysis, the isolated virus clustered with BoHV5 strains and showed highest similarity with the Brazilian BoHV5 strains.

## Introduction

Bovine herpesvirus types 1 (BoHV1) and 5 (BoHV5) belong to the family *Herpesviridae*, subfamily *Alphaherpesvirinae* and genus varicellovirus [1]. They are genetically and antigenically closely related and share ~85% nucleotide identity. BoHV1 is prevalent globally and causes infectious bovine rhinotracheitis (IBR). IBR is mainly a respiratory disease and is characterized by conjunctivitis and rhinotracheitis. Genital BoHV1 infection is characterized by balanoposthitis, infectious pustular vulvovaginitis and abortion. BoHV1 infection results in considerable economic losses due to decreased milk production, weight loss and abortions.

BoHV5 usually infects young calves and mortality can reach up to 100% [2]. BoHV5 mainly causes fatal meningoencephalitis in cattle and establishes latency in trigeminal ganglion. The virus is excreted in nasal, ocular and genital secretions upon reactivation (under stress). The clinical signs in affected cattle include depression, anorexia, weakness followed by neurological signs, such as incoordination, muscular tremor, blindness, circling, recumbency, head pressing, convulsions, paddling and death [3]. Occasionally, BoHV5 has been shown to be associated with the reproductive disorders [4].

**Funding:** This work was supported by the Science and Engineering Research Board, Department of

Science and Technology, Government of India, Grant number CRG/2018/004747 to N. Kumar and Indian Council of Agricultural Research, Grant Number IXX11882 to N. Kumar and SB. The funders had no role in study design, data collection and analysis, decision to publish, or preparation of the manuscript.

**Competing interests:** The authors have declared that no competing interests exist.

Herpesvirus associated bovine meningoencephalitis was first time reported in 1962 in Australia. Based on the virion morphology, cytopathic effect in cell culture and antigenic properties, the isolated virus was initially considered to be a neuropathogenic variant of BoHV1, called bovine encephalitis herpesvirus (BEHV) [5] or BoHV1 subtype 3 [1]. However, later on, based on the restriction site mapping of viral DNA and cross reactivity with monoclonal antibodies, BEHV was found to be a distinct strain with different genomic and antigenic properties. Thus, in 1992, International Committee on Taxonomy of Viruses recognised BEHV as a distinct virus species, namely BoHV5 [6].

The prevalence of BoHV5 is not precisely known because the available serologic tests do not discriminate antibodies against BoHV1- and BoHV5. Naturally occurring or vaccine-induced BoHV1 antibodies confer cross protection against BoHV5, a possible reason of non-occurrence of BoHV5-associated disease in BoHV1 endemic areas [7,8].

BoHV5 is endemic in South American countries-Argentina [9], Uruguay [10] and Brazil [11]. Only a few cases of the disease have been reported in Australia [12], Hungary [13], Iran [14], Canada [15] and United States [16]. BoHV5 has not been reported in India. We isolated and characterized BoHV5 for the first time from aborted cattle in India.

## Materials and methods

### Ethics statement

The study involves collection of biological specimens from cattle (field animals). Vaginal swabs and blood samples (3 ml each) were collected from three aborted and three apparently healthy cattle as per the standard practices without using anaesthesia. ICAR-National Research Centre on Equines, Hisar (India) granted the permission to collect the biological specimens. A due consent was also taken from the farmer (animal owner) before collection of the specimens.

### Cells and virus

Madin-Darby bovine kidney (MDBK) cells were grown in Dulbecco's Modified Eagle's Medium (DMEM) supplemented with antibiotics and 10% fetal calf serum. Reference BoHV1 (VTCCAVA14) was available in our repository at National Centre for Veterinary Type Cultures (NCVTC), Hisar, India which has been described elsewhere [17,18].

### Clinical specimens

The samples originated from an organized cattle farm located near Bhilwara, Rajasthan, India (25.3407˚ N, 74.6313˚ E). Out of a total of 68 animals, 61 (including 2 bulls) and 7 (including one bull) belonged to Gir and Tharpakar breeds, respectively. First evidence of abortion in the farm was noticed about 2 years prior to the sampling. Over 25 abortions had already occurred. Three cattle with a recent history of abortion (3–18 days prior to sampling) as well as 3 apparently healthy animals without any history of abortion were considered for sampling. Abortions occurred between 4–9 months of pregnancy without any specific pattern. No involvement of nervous system was recorded in any of the aborted cows. The farmer practiced natural service in the farm and never performed artificial insemination for breeding purpose. Besides six-monthly vaccinations against foot-and-mouth disease (FMD), the farmer also employed RB51 calfhood vaccination against *Brucella*. Bulls for breeding purpose were purchased from Surendranagar, Gujarat, a nearby state (India).

Vaginal swabs were collected in Minimum Essential Medium (MEM, transport medium) and transported on ice. The biological specimens were processed as per the standard procedures described elsewhere [19]. An aliquot of 500 µl suspension of vaginal swab was used for

bacteriological examinations. The remaining sample was filtered through 0.45 µM syringe filters and used for various virological assays. Serum/blood samples (3 ml from each animal) were also collected from aborted as well as healthy animals. A due consent was taken from the animal owner for collection of the biological specimens.

### Identification of the agent(s)

Initially, the DNA was extracted from vaginal swabs by DNeasy Blood & Tissue Kits (Qiagen, Valencia, CA, USA) and subjected to amplification of the BoHV1, *Campylobacter spp*, *Brucella spp*, *Leptospiran spp*, *Listeria spp* and *Trichomonas vaginalis* genomes, the agents which are commonly associated with abortion in cattle. Primers, annealing temperatures and extension times for amplification of these agents are given in Table 1. For PCR amplification, each reaction tube of 20 µl contained 10 µl of Q5 High-Fidelity 2× Master Mix (New England BioLabs Inc.), 20 pmol of forward and reverse primer, and 5 µl of DNA (template). Thermocycler conditions included: a denaturation step of 5 min at 98˚C followed by 35 cycles of amplification [(30 sec at 98˚C, 30 s at 58–65˚C (Table 1) and 40–90 s (Table 1) at 72˚C], and a final extension step at 72˚C for 10 min. The PCR products were separated in a 1% agarose gel.

**Table 1. Oligos to amplify various agents potentially involved in abortion.**

| Agent | Target Gene | Nucleotide sequences of the oligos | Product size (bp) | Thermocycler conditions | References |
|---|---|---|---|---|---|
| **BoHV1/5** | *UL27* (gB)[*] | Forward: 5'-CACGGACCTGGTGGACAAGAAG-3' | 484 | Annealing = 58˚C | [20] |
| | | Reverse: 5'-CTACCGTCACGTGAGTGGTACG-3' | | Extension = 40 s | |
| | *UL27* (gB)[#] | Forward: 5'– CGGCACGCTCGAACGGCAT –3' | 534 | Annealing = 58˚C | [21] |
| | | Reverse: 5'– AGCAGCTCGTTGTCCTCGC –3' | | Extension = 40 s | |
| | *UL44* (gC)[$] | BoHV5-Forward: 5'– CGGACGAGACGC CCT TGG –3' | 159 | Annealing = 58˚C | [22] |
| | | BoHV1-Forward: 5'– CAACCGAGACGGAAAGCTCC –3' | 354 | Extension = 30 s | |
| | | BoHV1/5-Reverse: 5'– AGTGCACGTACAGCGGCTCG –3' | | | |
| | *UL44* (gC)[$] | Forward: 5'– ATGGGCCCGCTGGGGCGAGC –3' | 1461 | Annealing = 65˚C | NA |
| | | Reverse: 5'– CTACAGGCGCGCCCGGGCCTTG –3' | | Extension = 90 s | |
| | *UL54* | Forward: 5'– TACTGCGCGCACTCGGGTAC –3' | 649 | Annealing = 58˚C | NA |
| | | Reverse: 5'–AGACGCTCATGGTCCACGGC –3' | | Extension = 40 s | |
| ***Brucella spp*** | *Omp2b* | Forward: 5'– GCGCTCAGGCTGCCGACGCAA –3' | 192 | Annealing = 58˚C | [23] |
| | | Reverse: 5'– ACCAGCCATTGCGGTCGGTA–3' | | Extension = 30 s | |
| ***Listeria spp*** | *hly* | Forward: 5'– CCTAAGACGCCAATCGAA –3' | 702 | Annealing = 50˚C | [24] |
| | | Reverse: 5'– AAGCGCTTGCAACTGCTC –3' | | Extension = 60 s | |
| ***Leptospira spp*** | *16s* | Forward: 5'– GGCGGCGCGTCTTAAACATG –3' | 329 | Annealing = 53˚C | [25] |
| | | Reverse: 5'– TCCCCCCATTGAGCAAGATT –3' | | Extension = 60 s | |
| ***Campylobacter spp*** | *Aspartate kinase* | Forward: 5'– GGTATGATTTCTACAAAGCGAG –3' | 503 | Annealing = 48˚C | [26] |
| | | Reverse: 5'– ATAAAAGACTATCGTCGCGTG –3' | | Extension = 60 s | |
| ***Trichomonas vaginalis*** | *B-tubulin* | Forward: 5'– CATTGATAACGAAGCTCTTTACGAT –3' | 112 | Annealing = 48˚C | [27] |
| | | Reverse: 5'– GCATGTTGTGCCGGACATAACCAT –3' | | Extension = 60 s | |

[*]Primers for initial diagnostic PCR,

[#]Primers for BoHV1/BoHV5 differential PCR,

[$] Primers for BoHV5 subtyping, s = seconds, bp = base pairs, NA = Not applicable (self-designed primers)

## Virus isolation

For virus isolation, aliquots of biological specimens (500 µl filtrate) were used to infect confluent monolayers of MDBK cells for 2 h followed by addition of fresh DMEM. The cells were observed daily for appearance of the cytopathic effect (CPE).

## Plaque assay

Plaque assay was performed as per the previously described methods along with some modifications [28,29]. The confluent monolayers of MDBK cells, grown in 6 well tissue culture plates, were infected with 10-fold serial dilutions of virus for 1 h at 37°C, after which the infecting medium was replaced with an agar-overlay containing equal volume of 2X L-15 medium and 2% agar. The plaques were visible at 5–6 days-post infection (dpi). The agar-overlay was removed, and the plaques were stained by 1% crystal violet. The viral titres were measured in plaque forming unit/ml (pfu/ml).

## Virus neutralization assay

Serum samples were first heated at 56°C for 30 min to inactivate the complements. MDBK cells were grown in 96 well tissue culture plates at ~90 confluency. Two-fold serum dilutions (in 50 µl volume) were made in phosphate buffer saline (PBS) and incubated with equal volume of either BoHV1 or BoHV5 ($10^3$ pfu/ml) for 1 h. Thereafter, virus-antibody mixture was used to infect MDBK cells. The cells were observed daily for the appearance of CPE. Final reading was taken at 72 hours post-infection (hpi) for the determination of antibody titers.

## Biotyping of BoHV5

Based on the restriction endonuclease digestion patterns, BoHV5 has 3 subtypes, *viz*; A, B and C [22,30]. We subjected BoHV5 to multiplex amplification of *UL27* and *UL54* genes by PCR followed by *BstE*II digestion. The banding pattern generated was used to determine biotypes of the BoHV5 as per the standard procedures [22,30].

## Nucleotide sequencing

In order to further confirm the identity of the virus, *UL27*, *UL44* and *UL54* genes were amplified by PCR, gel purified using QIAquick Gel Extraction Kit (Qiagen, Hilden, Germany) and subjected to direct sequencing using both forward and reverse PCR primers. Duplicate samples were submitted for sequencing and high-quality sequences were deposited in the GenBank database with an Accession Numbers viz; MN852441 (*UL27*), MN852442 (*UL44*) and MN852443 (*UL54*).

## Phylogenetic analysis

Nucleotide sequences from *UL27*, *UL44* and *UL54* genes (BoHV5//India/2018) were edited to yield 447, 1368 and 585 bp fragments respectively using BioEdit version 7.0. These sequences, together with the representative nucleotide sequences of BoHV1 and BoHV5 available in the public domain (GenBank) were subjected to multiple sequence alignments using CLUSTALW (http://www.ebi.ac.uk/clustalw/index.html). Phylogenetic analyses were conducted using MEGA X. [31]. In order to evaluate the evolutionary history of the strain as well as phylogenetic relationship with different lineages, a Neighbor-Joining concatermeric phylogenetic tree comprising of *UL27*, *UL44* and *UL54* genes was generated. Test of phylogeny was performed using Maximum Composite Likelihood method and the confidence interval was estimated by a bootstrap algorithm applying 1,000 iterations. Molecular phylogeny and genetic relatedness

of the isolated BoHV5, with the rest of the BoHV1/BoHV5 strains were calculated using % similarity and pairwise distances.

## Results

### Identification of the agent(s)

For demonstration of the etiological agents in the aborted cattle, we extracted DNA from the vaginal swabs and subjected for amplification of BoHV1, *Campylobacter spp*, *Brucella spp*, *Leptospira spp*, *Listeria spp* and *Trichomonas vaginalis* genomes, the agents which are commonly associated with abortion in cattle. Among the bacterial/parasitic agents, amplification could not be observed except for *Brucella spp* (data not shown). Amplification of a 484 nt long PCR fragment with BoHV1-specific oligos primarily indicated the presence of BoHV1 in vaginal swabs (Fig 1a). Among three aborted cattle, two were found positive for both BoHV1 and *Brucella* genomes whereas one was positive only for BoHV1 genome. The PCR product (BoHV1)

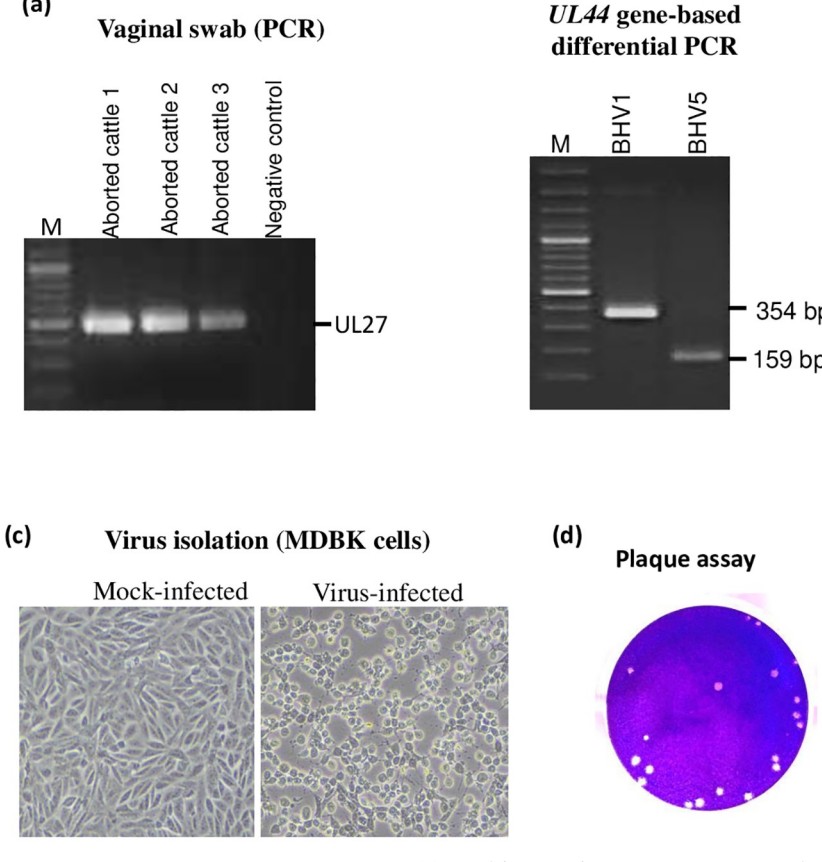

**Fig 1. Isolation and identification of the agent.** (a) *Amplification of BoHV1 genome in vaginal swab*. Virus was recovered from the vaginal swabs in DMEM followed by DNA extraction and PCR to amplify *UL27* gene of BoHV1. (b) *BoHV1/BoHV5 differential PCR*. PCR was carried out to amplify *UL44* gene as per the method described by Claus *et al*. PCR amplification of *UL44* from BoHV1 (reference) and BoHV5 (sample) resulted in amplification of 354 bp and 159 bp fragments respectively. (c). *Virus isolation in MDBK cells*. Virus recovered from the vaginal swab was used to infect MDBK cells. Cytopathic effect was observed at 3rd blind passage. Virus-infected and mock-infected cells are shown. (d) *Plaque assay*. Confluent monolayers of MDBK cells were infected with 10-fold serial dilutions of the virus for 1 h at 37°C followed by replacing the medium with an agar-overlay. The plaques were visible at 5–6 dpi. The agar-overlay was removed, and the plaques were stained by 1% crystal violet. The viral titers were measured as pfu/ml.

was subsequently gel purified and submitted for direct sequencing. Nucleotide sequence similarity using NCBI BLAST revealed a close homology with BoHV5 strains, rather than with BoHV1 strains. This finding was intriguing because BoHV5 had never been reported in India. When we re-examined the nucleotide sequences of the oligos [32] that were used to amplify *UL27* gene, the sequences were found to be conserved among BoHV1 and BoHV5 genome. Therefore, the oligos could amplify *UL27* gene of both BoHV1 and BoHV5.

The differential (BoHV1/BoHV5) PCR, based on *UL44* gene [21], resulted in amplification of 354 bp and 159 bp fragments, respectively from BoHV1 (reference) and BoHV5 (sample) (Fig 1b). This further confirmed that the virus detected in vaginal swab was BoHV5, not BoHV1.

## Virus isolation

The virus recovered from the vaginal swabs was used to infect MDBK cells, however no CPE was observed even up to 7 dpi. Thereafter, the infected cells were freeze-thawed twice and the resulting supernatant (called first blind passage) was used to reinfect fresh MDBK cells. In the third blind passage, the CPE, characterized by cell rounding, ballooning and degeneration was observed at ~36 hpi, in virus-infected but not in mock-infected cells (Fig 1c). Viral genome could be amplified in MDBK-infected cell culture supernatant at the $3^{rd}$ blind passage (data not shown). The isolated virus also formed plaques in MDBK cells which were visible within 5–6 days following infection (Fig 1d). Virus isolation was successful only in two out of the three vaginal swabs included in the investigation (Table 2). However, later, only one of the virus isolates was used for detailed analysis. The virus infected cell culture supernatant had a titre of ~$10^7$ pfu/ml at $3^{rd}$ blind passage. For bulk production, MDBK cells were infected at MOI of 0.1 of BoHV5 followed by virus harvest at 48 hpi by rapid freeze-thaw method. The bulk virus had a titre of $1.4*10^7$ pfu/ml. The virus was deposited in the microbial repository at NCVTC, Hisar, bearing an Accession Number VTCCAVA218 and named as BoHV5/*Bos taurus*-tc/India/2018/Bhilwara.

## Biotyping of BoHV5

Based on the restriction endonuclease pattern, BoHV5 has 3 subtypes, *viz*; A, B and C. By employing a previously described method, we also subjected the isolated BoHV5 for subtyping [22]. As per the method, multiplex PCR amplification of BoHV5 *UL27* and *UL54* genes (Fig 2a) and their subsequent digestion by *Bst*EII produces following banding patterns: (i) Type A: *UL27* (363 bp, 171 bp) and *UL54* (420 bp, 249 bp) (ii) Type B: *UL27* (534 bp) and *UL54* (420 bp, 249 bp) (iii) Type C: *UL27* (363 bp, 171 bp) and *UL54* (649 bp). In our study, PCR amplification of *UL27* and *UL54* genes and their subsequent digestion by *Bst*EII produced four DNA fragments viz; 420 bp, 363 bp, 249 bp and 171 bp (Fig 2b) which were suggestive of subtype A of BoHV5.

**Table 2. Virus isolation, detection of viral genome and antiviral antibodies in aborted and apparently health animals.**

| Animal ID | Abortion history | BoHV5 genome* | Virus isolation* | *Brucella* genome | Antibody titre | | |
|---|---|---|---|---|---|---|---|
| | | | | | (Anti-BoHV1) | (Anti-BoHV5) | *Brucella* |
| Mahima | Yes | (+) | (+) | (+) | 64 | 32 | (+) |
| Kishori | Yes | (+) | (+) | (-) | 32 | 32 | (+) |
| Jaya | Yes | (+) | (-) | (+) | 8 | 8 | (+) |
| Kavita | No | (-) | (-) | (-) | 16 | 32 | NT |
| Kirti | No | (-) | (-) | (-) | 64 | 8 | NT |
| Kapila | No | (-) | (-) | (-) | 128 | 128 | NT |

* = Vagnal swab, NT = Not tested, (+) = Positive, (-) = Negative

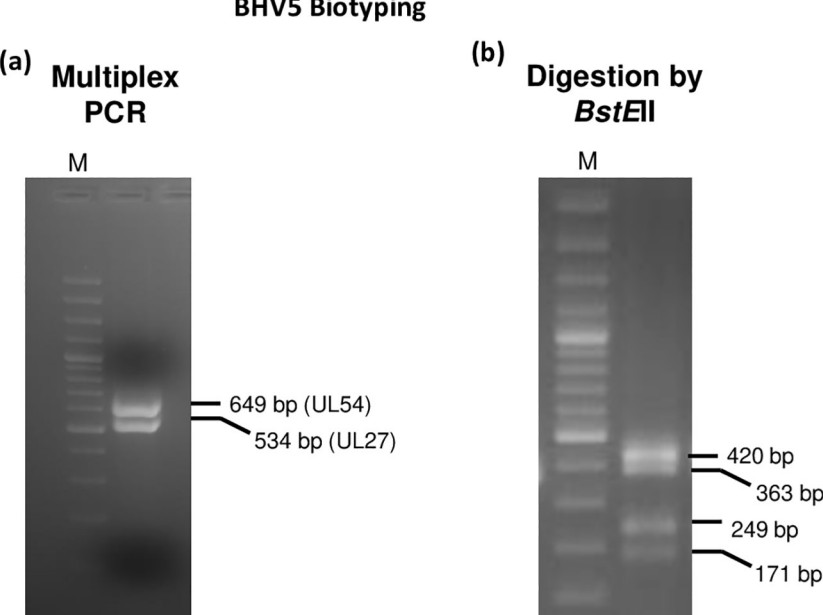

**Fig 2. BoHV5 subtyping. (a)** Amplification of *UL27* and *UL54* genes of BoHV5 by multiplex PCR resulted in amplification of 534 bp and 649 bp fragments respectively. **(b)** Digestion of PCR products (multiplex-*UL27* and *UL54* genes) by *BstE*II resulted in amplification of 420 bp, 363 bp, 249 bp and 171 bp fragments.

## Genetic relatedness

In UL44, the nucleotide (nt) and amino acid (aa) identities of the isolated virus with other BoHV5 strains were found to be in the range of 98.5–100% and 98.1–100% respectively. The highest identity (100%) was observed with a Brazilian isolate (KY549446.1). The nt and aa identities with BoHV1 isolates (*UL44*) were in the range of 91.6–96.7% and 90.0–95.9% respectively. In the *UL54* and *UL27*, the nt identities of the isolated virus with other BoHV5 strains ranged between 99.4–100% and 90.4–99.5% respectively.

## Phylogenetic analysis

Neighbour-joining phylogenetic tree comprising of *UL27*, *UL44* and *UL54* genes was constructed to ascertain the evolutionary relationship of the virus with other BoHV5 isolates in the public database. The Indian isolate clustered with the Brazilian BoHV5 isolates with 100% bootstrap suggesting that the virus is closely related with BoHV5 isolates from Brazil (Fig 3).

## Analysis of recombination

To elucidate any evidence of recombination between BoHV5 isolates, we also carried out recombination analysis in the *UL44* gene. The default parameters available in RDP4 programme *viz*; RDP, GENECONV, BOOTSCAN, MAXCHI, Chimera, SISCAN and TOPAL were employed to identify the recombination breakpoints as well as parental strains. However, the analysis did not reveal any evidence of recombination (data not shown).

## Detection of antiviral antibodies

We also evaluated the levels of antiviral antibodies in three aborted and three apparently healthy cattle belonging to the same farm. An antibody titre of 8–128 was observed irrespective

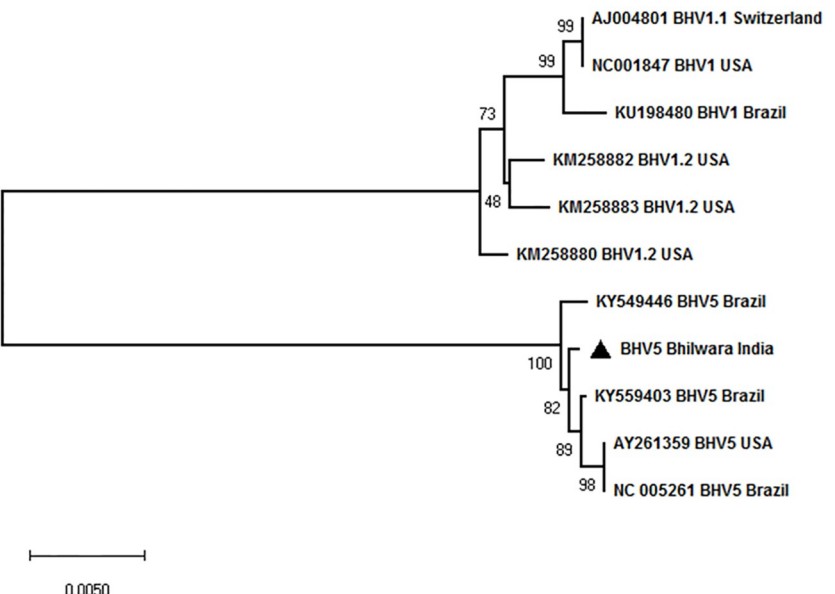

**Fig 3. Phylogenetic analysis.** Nucleotide sequences from *UL27*, *UL44* and *UL54* genes (BoHV5//India/2018) were edited to 447, 1368 and 585 bp fragments respectively, using BioEdit version 7.0. These sequences, together with the representative nucleotide sequences of BoHV1 and BoHV5 available in the public domain (GenBank) were subjected for multiple sequence alignments. Phylogenetic analyses were conducted using MEGA X. To evaluate the evolutionary history of the strain as well as the phylogenetic relationship with different lineages, a concatemeric Neighbour-Joining method tree was generated. Test of phylogeny was performed using Maximum Composite Likelihood method and the confidence intervals were estimated by a bootstrap algorithm applying 1,000 iterations. The tree is drawn to scale, with branch lengths in the same units as those of the evolutionary distances used to infer the phylogenetic tree.

of the abortion history (Table 2) suggesting an active BoHV5 infection in the herd. Serum samples of all the aborted cattle were also positive for antibodies against *Brucella* (Table 2).

## Discussion

BoHV5 distribution is restricted to South American countries, particularly Argentina [9], Uruguay [10] and Brazil [11]. Only a few cases of this disease have been reported from other countries [12,13,15,16]. BoHV5 has never been reported from India. Clinical findings, detection of antiviral antibodies, virus isolation, BoHV1/BoHV5 differential PCR, biotyping and sequence and phylogenetic analysis of *UL27*, *UL44* and *UL54* genes confirmed the association of BoHV5 subtype A in the aborted cattle. To the best of our knowledge this is the first report on the presence of BoHV5 infection in India.

DNA extracted from vaginal swabs of aborted cattle showed the presence of BoHV1 (three aborted cattle) and *Brucella* (2 aborted cattle). Other agents such as *Campylobacter spp*, *Listeria spp*, *Leptospira spp*, *Trichomonas vaginalis* were not detected. Rapid generation of CPE (within 36 hrs) in MDBK cells, a characteristic of BoHV1 further indicated association of BoHV1 in aborted cattle. However, nucleotide sequences revealed a close homology with the BoHV5 strains, rather than with the BoHV1 strains. When the corresponding nucleotide sequences of the primers [32] used to amplify *UL27* gene were re-examined, they were identical in BoHV1 and BoHV5 genomes. Therefore, the primer could amplify *UL27* gene of both BoHV1 and BoHV5 strains.

In order to further confirm, the isolated virus (BoHV5//India/2018/Bhilwara) and a reference BoHV1 control were subjected to differential PCR as described previously (Claus et al., 2005). As anticipated, PCR amplification of the *UL44* gene from BoHV1 and BoHV5 resulted

in amplification of 354 bp and 159 bp fragments, respectively, which further confirmed the identity of the virus as BoHV5, not BoHV1.

Based on the restriction endonuclease patterns [22,30], BoHV5 has 3 subtypes, *viz*; A, B and C. Type strains for subtypes A, B and non-A-non-B, are the Australian strain N569, the Argentinean strain A663 and Brazilian strains, respectively. The banding patterns generated following PCR amplification of *UL54* and *UL27* genes (BoHV5//India/2018/Bhilwara) and their subsequent digestion by *BstE*II were clearly suggestive of BoHV5 subtype A.

BoHV5 infection in cattle usually causes meninogoencephalitis [3], although few reports also suggest the involvement of reproductive tract [4]. Besides demonstration of the virus (BoHV5) and BoHV5-specific antibodies in the aborted animals, no obvious neurological signs could be recorded in any of the aborted cattle. There are reports suggesting interspecific recombination between BoHV1 and BoHV5 [33]. However our analysis did not reveal any evidence of recombination. Besides, we could not detect evidence of BoHV1 infection in the farm. Thus, BoHV5 may be a potential viral agent associated with abortion in cattle. Reproducing clinical BoHV5 disease under experimental conditions is a tedious task, although some laboratories have developed a rabbit model of encephalitits [34–38]. Although the isolated virus in our study was ~99% identical with the Brazilian BoHV5 strains, its complete genetic characterization (whole genome sequencing) as well as its ability to produce encephalitis in natural host and/or in rabbits needs to be elucidated which is beyond the scope of this manuscript.

Since vaccine (BoHV1/BoHV5) was never used in the farm, demonstration of antiviral antibodies, together with virus isolation, strongly suggested an active BoHV5 infection in the herd. However, latently infected cattle may also develop antiviral antibodies, with infection occurring sometime before the abortion. This needs further investigations. Furthermore, precise role of the isolated virus (BoHV5/India/2018/Bhilwara), other infectious agents (bacterial/parasitic) or their coinfections [39–43] needs to be examined.

BoHV1 vaccine induces cross-protection against BoHV5 disease in cattle [7]. Naturally occurring or vaccine-induced anti-BoHV1 antibodies are believed to reduce the occurrence of BoHV5-associated disease in BoHV1 endemic areas [44]. However, with reasons precisely unknown, this does not apply to the epidemiology and transmission of BoHV5 in South America. Even though Argentina and Brazil have a high percentage of BoHV1 seropositive cattle (24.8–84.1% and 19–85%, respectively) [reviewed in reference [3]], both countries have reported several cases of BoHV5 associated meningoencephalitis. Furthermore, serological tests which can differentiate anti-BoHV1 and anti-BoHV5 antibodies are not available. Thus, the actual prevalence of BoHV5 infection and hence economic significance remains unknown. Furthermore, the phenomenon of viral interference between BoHV1 and BoHV5 needs to be explored.

Like Europe and USA, since India has no specific programme for the detection and identification of BoHV5 infected animals, BoHV5 infection in India might have been overlooked despite its presence. The concerned farmer always practiced natural service and never performed artificial insemination for breeding purposes. However, the bulls were procured from nearby state (Surendranagar, Gujrat). India import semen from several other countries including Brazil, therefore the possibility of introduction of BoHV5 via semen from Brazil and/or other countries cannot be ruled out.

Very few studies have been undertaken on the development of BoHV5 vaccine, firstly because of its limited geographical distribution and secondly reproducing the clinical BoHV5 disease under experimental conditions is difficult [3,45]. Countries with frequent BoHV5 outbreaks along with high prevalence of BoHV1 have successfully employed BoHV1 vaccines for protection against BoHV5 infection [7,8]. However, the levels of anti-BoHV5 antibodies

induced by BoHV1 vaccine is usually low and of shorter duration [46]. Therefore, each BoHV1 vaccine should be carefully tested for potential cross-protection against BoHV5.

To conclude, we provide a strong evidence of BoHV5 infection in Indian cattle for the first time. The isolated virus would be useful for developing diagnostic, prophylactic and therapeutic agents to combat BoHV5 disease in India. The finding may necessitate inclusion of BoHV5 test protocol in testing of semen for sexually transmitted diseases.

## Supporting information

**S1 Raw Image.**
(PDF)

## Author Contributions

**Conceptualization:** Naveen Kumar.

**Data curation:** Naveen Kumar, Thachamvally Riyesh, Nitin Khandelwal, Ram Kumar.

**Formal analysis:** Naveen Kumar, Thachamvally Riyesh, Nitin Khandelwal, Ram Kumar.

**Funding acquisition:** Naveen Kumar.

**Investigation:** Naveen Kumar.

**Methodology:** Naveen Kumar, Yogesh Chander, Thachamvally Riyesh, Nitin Khandelwal, Ram Kumar, Harish Kumar.

**Project administration:** Naveen Kumar.

**Resources:** Naveen Kumar.

**Software:** Thachamvally Riyesh.

**Supervision:** Naveen Kumar.

**Validation:** Naveen Kumar, Yogesh Chander.

**Visualization:** Naveen Kumar.

**Writing – original draft:** Naveen Kumar.

**Writing – review & editing:** Naveen Kumar, Bhupendra N. Tripathi, Sanjay Barua.

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
