## [Decision Letter · Decision Letter 0]

1 Apr 2020

PONE-D-20-03577

Isolation and characterization of bovine herpes virus 5 (BHV5) from cattle in India

PLOS ONE

Dear Dr. Kumar,

Thank you for submitting your manuscript to PLOS ONE. After careful consideration, we feel that it has merit but does not fully meet PLOS ONE’s publication criteria as it currently stands. Therefore, we invite you to submit a revised version of the manuscript that addresses the points raised during the review process.

Please follow all the comments and suggestions from the reviewer 1 that will largely improve the soundness of this article.

We would appreciate receiving your revised manuscript by May 16 2020 11:59PM. To enhance the reproducibility of your results, we recommend that if applicable you deposit your laboratory protocols in protocols.io, where a protocol can be assigned its own identifier (DOI) such that it can be cited independently in the future. For instructions see: http://journals.plos.org/plosone/s/submission-guidelines#loc-laboratory-protocols

We look forward to receiving your revised manuscript.

Kind regards,

Pierre Roques, Ph.D.

Academic Editor

PLOS ONE

Journal Requirements:

Additional Editor Comments (if provided):

Sorry for the delay

Reviewers' comments:

Reviewer's Responses to Questions

**Comments to the Author**

1. Is the manuscript technically sound, and do the data support the conclusions?

Reviewer #1: Partly

Reviewer #2: Yes

2. Has the statistical analysis been performed appropriately and rigorously? 

Reviewer #1: N/A

Reviewer #2: Yes

3. Have the authors made all data underlying the findings in their manuscript fully available?

Reviewer #1: Yes

Reviewer #2: Yes

4. Is the manuscript presented in an intelligible fashion and written in standard English?

Reviewer #1: Yes

Reviewer #2: Yes

5. Review Comments to the Author

Reviewer #1: The manuscript is really interesting since there are only few reports associating BoHV-5 with abortion in cattle. Thus, this finding is novel and extends the geographical distribution of the virus to regions were infections have not been frequently reported. Nevertheless, there are some major points that the authors should address to better understand the implication of BoHV-5 in the case reported.

It is suggested to use the current virus nomenclature abbreviation for referring to the bovine alphaherpesviruses….BoHV-1 and BoHV-5

Virus seroneutralization test. …The cells were observed daily for appearance of CPE for determination of the antibody titres….

The authors should indicate whether they followed the standard seroneutralization procedure in which the test is read at 72 h…..it is not clear whether the read out was performed at the time of CPE appearance or when indicated by the control back titration at 72 h

Virus isolation. It is not clear from how many vaginal swabs the virus was isolated. It would be interesting to add a column in table 2 indicating isolation +or -

This point is important since Brucella is also a cause of abortion in cattle and only detection of BoHV5 genome is not indicative of virus involvement in the abortions.

Anti BoHV antibodies. According to the manuscript cattle in this study is not vaccinated against BoHV. However, since seroconversion is not demonstrated the presence of antibodies might indicate that cattle is latently infected, with infection occurring some time before the abortions. This should be indicated in the discussion.

Minor comments

Line 21. amplification of the viral genome in PCR….change IN PCR for BY PCR

Line 38. Bovine herpesvirus types -----Please change by current ICTV nomenclature…..bovine alphaherpesvirus types…

The following paragraph from lines 40 to 43 should be re-written. ….since IBR is the main respiratory disease it is characterized by conjunctivitis and rhinotracheitis….the genital infection is characterized by balanopostitis and infectious pustular vulvovaginitis and abortion ……..

40BHV1 is prevalent globally

41 and causes infectious bovine rhinotracheitis (IBR) in cattle which is characterized by

42 conjunctivitis, rhinotracheitis, balanopostitis, infectious pustular vulvovaginitis, enteritis,

43 abortion and sometimes encephalitis

Line 45….and mortality can causes up to 100%.....please change….and mortality can reach 100%

Line 171. ..been reported from India. Please change to reported IN India

Reviewer #2: This is a well-written manuscript that presents new information about the presence of BHV-5 in India, and of its potential to cause abortions. This manuscript should be of interest to those virologist that are presently working with this virus.

6. PLOS authors have the option to publish the peer review history of their article (what does this mean?). If published, this will include your full peer review and any attached files.

Reviewer #1: No

Reviewer #2: No

---

## [Author Response · Author response to Decision Letter 0]

6 Apr 2020

Reviewers response

Reviewer #1: The manuscript is really interesting since there are only few reports associating BoHV-5 with abortion in cattle. Thus, this finding is novel and extends the geographical distribution of the virus to regions were infections have not been frequently reported. Nevertheless, there are some major points that the authors should address to better understand the implication of BoHV-5 in the case reported.

We thank the Reviewer for his/her enthusiasm towards this study

It is suggested to use the current virus nomenclature abbreviation for referring to the bovine alphaherpesviruses….BoHV-1 and BoHV-5

BHV1/BHV5 has been changed to BoHV1/BoHV5 in the entire manuscript

Virus seroneutralization test. …The cells were observed daily for appearance of CPE for determination of the antibody titres….

The authors should indicate whether they followed the standard seroneutralization procedure in which the test is read at 72 h…..it is not clear whether the read out was performed at the time of CPE appearance or when indicated by the control back titration at 72 h

The final readings were taken at 72 h-post infection. The same has been described in the revised manuscript (materials and methods section).

Virus isolation. It is not clear from how many vaginal swabs the virus was isolated. It would be interesting to add a column in table 2 indicating isolation +or -

This point is important since Brucella is also a cause of abortion in cattle and only detection of BoHV5 genome is not indicative of virus involvement in the abortions.

Anti BoHV antibodies. According to the manuscript cattle in this study is not vaccinated against BoHV. However, since seroconversion is not demonstrated the presence of antibodies might indicate that cattle is latently infected, with infection occurring some time before the abortions. This should be indicated in the discussion.

Virus isolation was successful only in two out of three vaginal swabs. The same has been mentioned in the text (results) as well as in Table 2 in the revised manuscript.

Minor comments

Line 21. amplification of the viral genome in PCR….change IN PCR for BY PCR

Changed as per the suggestion

Line 38. Bovine herpesvirus types -----Please change by current ICTV nomenclature…..bovine alphaherpesvirus types…

BHV1/BHV5 has been changed to BoHV1/BoHV5 in the entire manuscript

The following paragraph from lines 40 to 43 should be re-written. ….since IBR is the main respiratory disease it is characterized by conjunctivitis and rhinotracheitis….the genital infection is characterized by balanopostitis and infectious pustular vulvovaginitis and abortion ……..

40BHV1 is prevalent globally

41 and causes infectious bovine rhinotracheitis (IBR) in cattle which is characterized by

42 conjunctivitis, rhinotracheitis, balanopostitis, infectious pustular vulvovaginitis, enteritis,

43 abortion and sometimes encephalitis

Rewritten as per the suggestions

Line 45….and mortality can causes up to 100%.....please change….and mortality can reach 100%

Changed as per the suggestion

Line 171. ..been reported from India. Please change to reported IN India

Changed as per the suggestion

Reviewer #2: 

This is a well-written manuscript that presents new information about the presence of BHV-5 in India, and of its potential to cause abortions. This manuscript should be of interest to those virologist that are presently working with this virus.

We thank the reviewer for his/her enthusiasm towards this study.

---

## [Editor Report · Decision Letter 1]

8 Apr 2020

Isolation and characterization of bovine herpes virus 5 (BoHV5) from cattle in India

PONE-D-20-03577R1

Dear Dr. Kumar,

We are pleased to inform you that your manuscript has been judged scientifically suitable for publication and will be formally accepted for publication once it complies with all outstanding technical requirements.

With kind regards,

Pierre Roques, Ph.D.

Academic Editor

PLOS ONE
---

## [Editor Report · Acceptance letter]

10 Apr 2020

PONE-D-20-03577R1 

Isolation and characterization of bovine herpes virus 5 (BoHV5) from cattle in India 

Dear Dr. Kumar:

I am pleased to inform you that your manuscript has been deemed suitable for publication in PLOS ONE. Congratulations! Your manuscript is now with our production department. 

With kind regards,

on behalf of

Dr. Pierre Roques 

Academic Editor

PLOS ONE